# In Vitro Evaluation of Biological and Anticancer Activities Exhibited by Five Varieties of *Vitis vinifera* L.

**DOI:** 10.3390/ijms241713440

**Published:** 2023-08-30

**Authors:** Linda Darwiche, Joelle Mesmar, Elias Baydoun, Walid El Kayal

**Affiliations:** 1Faculty of Agricultural and Food Sciences, American University of Beirut, P.O. Box 11-0236, Beirut 1107, Lebanon; 2Faculty of Arts and Sciences, American University of Beirut, P.O. Box 11-0236, Beirut 1107, Lebanon

**Keywords:** pancreatic cancer, breast cancer, cancer, *Vitis vinifera*, anticancer, phytochemical analysis

## Abstract

*Vitis vinifera* commonly known as grapevine is one of the most important fruit crops worldwide. Its cultivation started more than 7000 years ago in the Near East, and over the millennia was followed by the development of thousands of cultivars that were further selected and characterized for specific purposes. Its important pharmacological value and its richness in phytoconstituents were the triggers to perform this project. Seven extracts were prepared from five different *V. vinifera* varieties (*V. vinifera* ‘Black Pearl’ (BP), *V. vinifera* ‘Red Glob’ (RG), *V. vinifera* ‘Crimson’ (CR), *V. vinifera* ‘Beitamouni’ (BE) and *V. vinifera* ‘Superior’ (SU)) by separating the pulp from the seeds, followed by methanolic extraction. The phytochemical analysis showed that red colored grapes (RE, BP and CR), the seeds from *V. vinifera* ‘Black Pearl’ and *V. vinifera* ‘Red Globe’ contain higher amounts of primary and secondary metabolites such as polyphenols, anthocyanins and reducing sugars. In addition to their richness in phytoconstituents, these varieties/seeds possess an important antioxidant activity. The results of the cell viability assays showed that the red varieties have a potential anticancer activity against Capan-2 pancreatic cancer and MDA-MB231(TNBC) breast cancer cell lines, with the greatest promise when combined with the seeds.

## 1. Introduction

Grapevine, biologically known as *Vitis vinifera* L., is responsible for the production of the world’s third most valuable horticultural crop, namely grapes. Based on the hectares cultivated and economic value, grapevine is globally considered as one of the major fruit crops [1]. Although nowadays grapes are grown everywhere, they are still mostly cultivated in places with a temperate, Mediterranean-style climate [2]. Despite the existence of between 5000 and 10,000 varieties of *Vitis vinifera*, only a few have commercial significance for both wine and table grape production; however, these varieties still account for nearly 90% of cultivated grapes worldwide [3,4].

This fruit’s high variability in shapes, colors, sizes and taste, along with many other physical and chemical differences, has caught the attention of biologists and phytotherapists, promoting them to investigate its pharmacological importance. Herbal medicine emerged a long time ago, and to this day, more than 80% of the world’s population still relies on herbal medicine due to various reasons such as culture, religious beliefs, plant availability, affordability, rare side-effects, autonomy and ease of use [5,6]. About 25% of the drugs prescribed worldwide are of plant origin, and these extracts are used in the treatment of both chronic and acute conditions such as cardiovascular disease, prostate problems, depression and inflammation, as well as an immune system booster [7].

According to the WHO, cancer is one of the most chronic diseases worldwide and a leading cause of death, accounting for more than 10 million deaths in 2020 (1 in 6 deaths) [8,9]. A review study indicates that over 70% of the world’s population is incapable of affording cancer treatment expenses, and in most cases, cancer therapy approaches exhibit minimal efficacy with elevated levels of intolerable toxicity [10]. Due to the aforementioned advantages of herbal medicine and the fact that plants are rich in primary and secondary metabolites, scientists have been keen on preparing plant extracts and testing them on different cancer cell lines, with promising results [11].

Pancreatic cancer is the number one asymptomatic cancer, with adenocarcinoma being the most common type. Despite the application of some chemotherapeutic drugs to treat pancreatic cancer patients, drug resistance still leads to poor clinical outcomes and low survival rates. Consequently, pancreatic cancer ranks among the leading causes of cancer-related deaths, necessitating the urgent development of alternative treatment therapies and novel potential therapeutic drugs [12].

Breast cancer is not only the most common type of cancer, but also the leading cause of cancer-related death in women worldwide. Despite the presence of a variety of therapy modes and the emergence of new ones, both the incidence of, and death from breast cancer have increased over the last three decades [13]. The most lethal subtype of breast cancer is triple-negative breast cancer (TNBC), due to its high heterogeneity, aggressive nature and lack of treatment options. Chemotherapy is the treatment of choice for TNBC; however, unfortunately, patients frequently develop resistance [14]. Therefore, efforts in recent years have been focused on finding alternative treatments using natural sources with the goal of limiting resistance.

The *Vitis vinifera* plant is very rich in a variety of phytoconstituents, including phenolic compounds, flavonoids and anthocyanins. Its medical importance lies in its skin protection, antioxidant, antibacterial, anticancer, anti-inflammatory and antidiabetic activities, as well as its hepatoprotective, cardioprotective and neuroprotective effects [15]. Studies have shown that grape seed extracts (GSE) possess potential anticancer activity against oral squamous cell carcinoma (KB cells) [16], human colorectal cancer (CRC cell) [17], and chemo-resistant ovarian cancer (OVCAR-3 cells) [18]. Additionally, grape stem extracts showed anticancer activity against colon (HT29), breast (MCF-7 and MDA-MB-23), renal (786-0 and Caki-1) and thyroid (K1) cancer cells [19].

Although the anticancer effects of grape seed and stem extracts have been tested on different cancer cell lines, the anticancer effect of the whole fruit extract has been scarcely studied. Thus, in this study, five *V. vinifera* varieties grown in different locations in Lebanon were used to prepare seven extracts. Three of the varieties are seedless, namely *V. vinifera* ‘Superior’ (SU), *V. vinifera* ‘Beitamouni’ (BE), which is a local crop in Lebanon, and *V. vinifera* ‘Crimson’ (CR). The other two varieties are seeded, namely *V. vinifera* ‘Black Pearl’ (BP) and *V. vinifera* ‘Red Globe’ (RE). The seeds were separated and extracted from the fruits in the seeded varieties.

## 2. Results

### 2.1. Qualitative Phytochemical Analysis for VVM Extracts

Bioactive compounds are produced by plants in small amounts and have extra-nutritional roles [20]. The phytochemical investigation performed in our lab showed the presence of various phytochemical substances in *Vitis vinifera* methanolic (VVM) extracts, as shown in Table 1. Tannins and saponins were only found in BPS and RGS, whereas resins were only found in RGS. Phenols, quinones, terpenoids, anthocyanins and reducing sugars were found in all of the seven extracts. No anthraquinones or steroids were detected in the tested extracts. In addition, cardiac glycosides were found in CR, BP and BPS.

### 2.2. Quantitative Phytochemical Analysis of VVM Pulp and Seed Extracts

As shown in Figure 1, only the seeds from BP and RG contain high level of polyphenols, with that of RGS with the highest value (0.333 ± 0.012 μg GA/mg dry extract). Total phenolic content in the flesh extracts ranged from 0.004 ± 8.4 × 10^−5^ to 0.007 ± 11 × 10^−5^ μg GA/mg.

The quantification of anthocyanins in the RG, RGS, BE, BPS, BPS, SU and CR extracts was performed, where the anthocyanin concentration of the sample was expressed as cyanidin-6-O-glucoside equivalent (mg C6GE/L), as shown in Figure 2. Since anthocyanins are purple pigments, their levels were higher in the red varieties (CR, BP and RG) with negligible levels in the green ones. The cyanidin-6-O-glucoside equivalent levels was the highest in BP (81.71 ± 3.44 mg C6G/L), which was almost 8-fold greater than that of CR (7.46 ± 0.95 mg C6G/L) and RG (13.30 ± 0.20 mg C6G/L) as shown in Figure 2.

As part of the quantitative phytochemical analysis, reducing sugar content (RSC) was quantified by measuring glucose equivalence per gram dry weight of extract (mg G/g dry weight). The results showed that BPS extract was the richest in reducing sugars with 963.11 ± 19.88 mg G/g dry weight content, which was almost 2 times that of BP which was 539.62 ± 4.52 mg G/g dry weight (Figure 3).

### 2.3. VVM Extracts Have Variable Antioxidant Activities

*V. vinifera* has been reported to be rich in many phytochemical constituents [21,22] and this was confirmed using phytochemical analysis. The antioxidant capacity of the seven extracts was assessed in vitro using the DPPH-radical scavenging assay. DPPH is a stable free radical that antioxidants react with and convert to α, α-diphenyl-β-picryl hydrazine, thereby changing the color of the solution from purple to a pale yellow. Our results showed that both seed extracts (BPS and RGS) exhibited significant free radical-scavenging activity in a concentration-dependent manner, with RGS showing the highest activity (80% radical scavenging activity). Their pulp counterparts had similar antioxidant capacities that were higher than that exhibited by the seedless varieties as shown in Figure 4.

### 2.4. VVM Extracts Inhibit the Proliferation of Capan-2 and MDA-MB231 Cancer Cells

The grape pulp and seed extracts were tested for their anticancer activity against two aggressive cancer cell lines. For this purpose, we examined the anti-proliferative effect of various concentrations (0, 50, 100, 200, 400 and 600 μg/mL) of the extracts against the MDA-MB-231 triple negative breast cancer cell line and Capan-2 pancreatic adenocarcinoma cell line. Results showed that the CR pulp extract exhibited the strongest activity among the seedless varieties, particularly against the MDA-MB-231 cells (Figure 5). Moreover, the BPS extract exhibited the highest anti-proliferative potential against the breast and capan-2 cancer cell lines used, in concentration and time-dependent manners, compared to other grape seed extracts RGS (Figure 6)**.** Overall, the seed extracts (BPS and RGS) show a much stronger anticancer activity compared to the pulp extracts (SU, CR, BE, BP, and RG) (Figure 5 and Figure 6). This is in alignment with their antioxidant activity and phytochemical profiles, described above.

## 3. Discussion

In this study, five different cultivars of *V. vinifera* were used: three seedless varieties (SU, CR, BE) and two seeded (RG, BP). Additionally, the varieties differ in their colors, with three red varieties (CR, BP, RG) and two white ones (SU&BE). To minimize bias and compare the cytotoxic effect of seedless varieties to seeded ones, the seeds of the seeded varieties were separated from the pulp, allowing us to compare the effects between pulp extracts of the seeded and seedless varieties, and between seed and pulp extracts. Phytochemical analysis showed that only seeds (BPS & RGS) contain saponins and tannins. Tannic acid has been shown to have a role in the induction of cell cycle arrest, apoptosis and limiting the proliferation of various cancer cells [23]. Additionally, saponins have shown effective anticancer potential in various cancer cell lines by aiding in the inhibition of cell growth and induction of apoptosis [24]. Only RGS contains resins, which also induces apoptosis in human cancer cells [25]. It was also found that all of the seven extracts contain phenols, quinones, terpenoids, anthocyanins and reducing sugars. Quinones are secondary plant metabolites that have both anti-proliferative and anti-metastatic roles when tested against various cancer types both in vitro and in vivo [26]. Some terpenoids have the ability to trigger certain stages of cancer progression through different mechanisms, such as suppressing the early stage of tumorigenesis [27]. Furthermore, cardiac glycosides (CG) were detected in CR, BP and BPS, which, recent studies suggest, induce the immunogenic attack of cancer cells [28]. After measuring the levels of anthocyanins in RG, RGS, BP, BPS, SU, BE and CR, it was found that Black pearl extract has the highest cyanidin-6-O-glucoside equivalent levels and, thus, the highest anthocyanin levels. This was followed by extracts from Red Globe and Crimson, the red varieties. The levels of anthocyanins in the seeds were similar, and levels in the white varieties were almost negligible, suggesting the absence of anthocyanin biosynthesis. The level of anthocyanins was the highest in the red varieties and very low in the white ones, validating the theory of VvMYBA1 mutation. This experiment was consistent with the antioxidant activities measured by DPPH, where red cultivars possessed higher antioxidant activity compared to the white ones, except for CR, whose radical scavenging potential was similar to that of the local Lebanese variety BE and that of SU. The antioxidant activity and total phenol content were the highest in the seed extracts. The production of polyphenols requires reducing sugars as building blocks; as seen by the significant amounts present in the seeds when measuring the RSC. Based on the above testing, the anti-proliferative potential of the seven extracts against MDA-MB-231 and Capan-2 cells showed that all the extracts have potential anticancer activity, but the most effective treatment was that of the seeds, with BPS having the highest potential. From the quantification of the total polyphenol content in the seven prepared extracts, it was found that the seeds contain the highest levels of gallic acid, which is a total polyphenol representative; RGS has the highest levels, followed by BPS, and the fruit pulp has very low levels.

Sugar plays a vital role in plants by being a part of their nutritive molecules and central signaling or regulatory molecules, thus, modulating gene expression related to plant growth, development, metabolism, stress response and disease resistance [29,30]. The reducing sugar content in a plant varies depending on various factors, such as genotype, age of the plant, soil quality, geographical location, climatic conditions, cultivation method and abiotic stress [31]. Quantification of reducing sugar content in grapes provides a crucial parameter to determine the alcoholic level during grape wine production and to check the glucide level during the fermentation process [32]. Using the DNSA method, the reducing sugar content was determined in the seven extracts, and the results showed that BPS extract has the highest glucose equivalent level, followed by RG, CR, RGS, BE, SU and BP. The synthesis of plant phenolic compounds is achieved through two basic pathways, the shikimic acid pathway and the acetate-malonate (polyketide) pathway. Erythrose-4-phosphate, which is the product of the pentose phosphate pathway, and phosphoenolpyruvate, a product of glycolysis, are the substrates required in the shikimic acid pathway. The mentioned pathways utilize glucose as the first crucial substrate [33]. The human body and its organs require a certain amount of free radicals for proper functioning, including reactive oxygen species (ROS) and reactive nitrogen species. Redox homeostasis is responsible for keeping these radicals in balance. In some cases, an off-balance situation causes an unexpected increase in these free radicals, leading to oxidative stress that can result in aging and chronic degenerative diseases, such as coronary heart disease and cancer [34]. The prepared extracts were assayed for their capacity to neutralize free radicals, since they are rich in polyphenols, anthocyanins and many other secondary metabolites that have proven antioxidant activities. Polyphenols have the ability to reduce oxidative stress either directly, by preventing free radical formation, or indirectly, by increasing the activity of key antioxidant enzymes [35]. Also, anthocyanins have the potential to prevent or inhibit oxidation by scavenging ROS and reducing oxidative stress [34]. The DPPH results showed that RGS had the highest radical scavenging activity, followed by BPS and then, the pulp of the seeded varieties (RG and BP). Those with the lowest radical scavenging activity were CR, BE and SU.

These results were consistent with those of the total phenol content (TPC) and the total reducing sugar content (RSC), where RGS had the highest TPC, the highest radical scavenging activity, and a significant RSC. Moreover, BPS had high RSC, thus, high levels of TPC, and a significant radical scavenging activity. The level of anthocyanins was low in the seeds asthe purple pigment is found in the fruit’s skin. As *V. vinifera* cultivars are very rich in different phytoconstituents, studies have been conducted to test their anti-proliferative and cytotoxic potential against different cell lines, such as A549, B164A5, HL-60, MCF-7, HT-29 and HeLa [21,36]. The anti-proliferative effects of the cultivars’ pulp and seed extracts were tested against MDA-MB-231 and Capan-2 cell lines. Among the seedless varieties, CR possessed the most prominent anti-proliferative activity against both cell lines. Comparing the pulp extracts to their respective seed extracts, it was found that seed extracts have a higher and more significant anti-proliferative activity, with BPS showing the most cytotoxic effect against both cell lines. Principal component analysis revealed important correlations between the quantified secondary metabolites and the antioxidant level and cell viability. The directly proportional correlation between TRS, TPC and anthocyanins is due to the fact that anthocyanins contain sugar moieties, mostly as 3-O-glycoside forms of arabinose, glucose, rhamnose and galactose in their pyranoside forms [37], and polyphenols need sugar to be produced [33]. In addition, the negative correlation between cell viability and DPPH results is due to antioxidant activity, which is accompanied by a decrease in cell viability. The more potent the treatment is, the higher is its ability to perform its cytotoxic activity against the cancer cells.

## 4. Materials and Methods

### 4.1. Preparation of Vitis vinifera Extracts

*Vitis vinifera* varieties were collected from six locations at different elevations above sea level, and the seeds of the *V. vinifera* ‘Red Globe’ and *V. vinifera* ‘Black Pearl’ varieties were separated from the skin. The pulp and seed stocks were frozen in liquid nitrogen and then stored at −80 °C until they were ready to be used. Subsequently, they were powdered using a food processor and lyophilized under conditions of low temperature and high pressure. The resulting residue was suspended in 80% methanol and incubated at room temperature for three days with constant shaking. Afterward, the suspension was filtered, dried using a rotary vacuum evaporator and lyophilized. The obtained powders were dissolved in DMSO at a concentration of 200 mg/mL and stored at −20 °C until further use.

### 4.2. Phytochemical Tests

Qualitative tests to detect the presence of secondary metabolites in the pulp and seed extract were carried out. The addition of certain reagents can cause a chemical reaction; consequently, changes in the color of the solution or the production of gas or precipitate were monitored. The tests performed, including the added reagent and the expected result, are shown in Table 2.

### 4.3. Total Phenolic Content (TPC)

The total phenolic content (TPC) of the *Vitis vinifera* pulp and seed extracts were determined by the Folin–Ciocalteu colorimetric method, using the BQC KB03006 polyphenol quantification assay kit (BQC Redox Technologies, Asturias, Spain) following the manufacturer’s protocol. Gallic acid was used as the reference standard and results were expressed as µg gallic acid equivalent (GAE) per mg of sample.

### 4.4. Total Anthocyanin Content (TAC)

The total anthocyanin content (TAC) in the *Vitis vinifera* pulp and seed extracts was quantified using the BQCkit anthocyanins assay kit (BQC Redox Technologies, Asturias, Spain) according to the manufacturer’s protocol. The anthocyanin concentration of the sample was expressed as cyanidin-6-O-glucoside equivalent (mg C6GE/L).

### 4.5. Total Reducing Sugar Content (RSC)

The 3,5-dinitrosalicyclic acid (DNS) method was used to estimate the reducing sugar content (RSC) in the pulp and seed extracts. For each variety, 1 mL of 1 mg/mL concentration was mixed with 1 mL DNS solution and placed in a boiling water bath for 10 min. The mixture was then cold shocked by placing the tubes in ice water. After cooling, 5 mL of distilled water was added to each sample and the absorbance of the solution was read at 540 nm. The blank was prepared by mixing 1 mL of DMSO water with 1 mL of DNS and then treated as the rest of the samples. The experiment was repeated three times. Absolute glucose was used to plot the standard curve. The RSC was expressed as mg glucose per gram of extract (mg G/g).

### 4.6. Assay of Antioxidant Activity

The antioxidant activity of the various grape extracts was tested using the radical scavenging activity of α, α-diphenyl-β-picrylhydrazyl (DPPH; Sigma-Aldrich Co., St. Louis, MO, USA) as previously described, with some modifications [38]. Volumes of 0.5 mL of different concentrations of the extracts (50, 100, 200, 400 and 600 µg/mL) were mixed with 0.5 mL of DPPH solution (0.5 nM in methanol) and 3 mL methanol. The blank consisted of 0.5 mL of DMSO with 0.5 mL of DPPH and 3 mL of methanol. The samples were then incubated in the dark at room temperature for 30 min, after which, the absorbance was measured at 517 nm using a spectrophotometer. The percentage of radical scavenging activity of each concentration of the extracts was determined using the formula
$$(\frac{\left(OD\;blank-OD\;plant\;extract\;at\;each\;concentration\right)}{OD\;Blank})*100.$$

### 4.7. Culture of MDA-MB-231 and Capan-2 Cell Lines

MDA-MB-231 human breast cancer cells (American Type Culture Collection, Manassas, VA, USA) and Capan-2 human pancreatic cancer cells (CLS Cell Line Service, Eppelheim, Germany) were cultured in DMEM high glucose medium supplemented with 10% fetal bovine serum (FBS) (both from Sigma-Aldrich, St. Louis, MO, USA) and 1% penicillin/streptomycin (Lonza, Switzerland). Cells were maintained in a humidified incubator (37 degrees Celsius and 5% CO_2_).

### 4.8. Assay of Cell Viability

MDA-MB-231 and Capan-2 cells (5 × 10^3^) were seeded in 96-well plates and allowed to grow until they reached 30–40% confluence. Cells were then treated with increasing concentrations of the indicated extracts (50 µ/mL, 100 µ/mL, 200 µ/mL, 400 µ/mL and 600 µ/mL) and incubated for a total period of 72 h. The viability of the cells was determined by the reduction of 3-(4,5-dimethylthiazol-2-yl)-2,5-diphenyltetrazolium bromide (MTT; Sigma-Aldrich, St. Louis, MO, USA) and calculated as the proportional viability of the treated cells compared to the DMSO vehicle-treated cells. Assays were performed in triplicate and repeated three times. Data are represented as mean values ± SEM.

### 4.9. Statistical Analysis

All experiments were carried out in triplicates and repeated three times. Data were represented as mean values ± SEM. The data were statistically evaluated using Student’s *t*-test using GraphPad Prism version 5.0. For the comparison of more than two means, one-way ANOVA (Dunnett’s post hoc test). Data were presented as mean ± standard error of the mean (SEM). A *p*-value of less than 0.05 was considered as significant, and (* denotes *p* < 0.05, ** denotes a *p* < 0.005, *** denotes *p* < 0.001 and **** denotes *p* < 0.0001).

## 5. Conclusions

In conclusion, our results and analysis show that *V. vinifera* cultivars inhibit the proliferation of Capan-2 and MDA-MB231 cancer cell lines, with the strongest effect conferred by the seeds, particularly those of BPS. Thus, based on the above findings, *V. vinifera* is a potential source for a therapeutic anti-cancer drug. However, further research is still needed to investigate the efficiency of *V. vinifera* in vivo and in vitro.

## Figures and Tables

**Figure 1 ijms-24-13440-f001:**
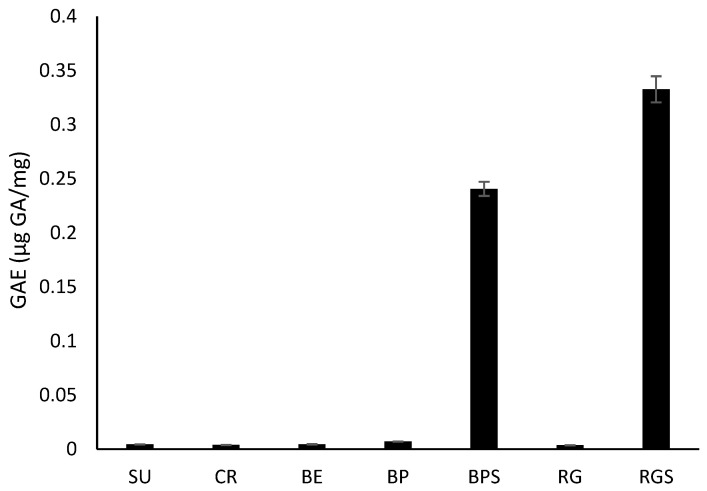
Spectrophotometric determination of the total phenolic content of VVM pulp and seed extracts. The graph shows the TPC in the 7 VVM extracts where it is expressed as µg Gallic acid equivalents (GAE) per milligram of sample in dry weight (µg/mg). The experiment was carried out in triplicates and repeated three times. Data are represented as mean values ± SEM.

**Figure 2 ijms-24-13440-f002:**
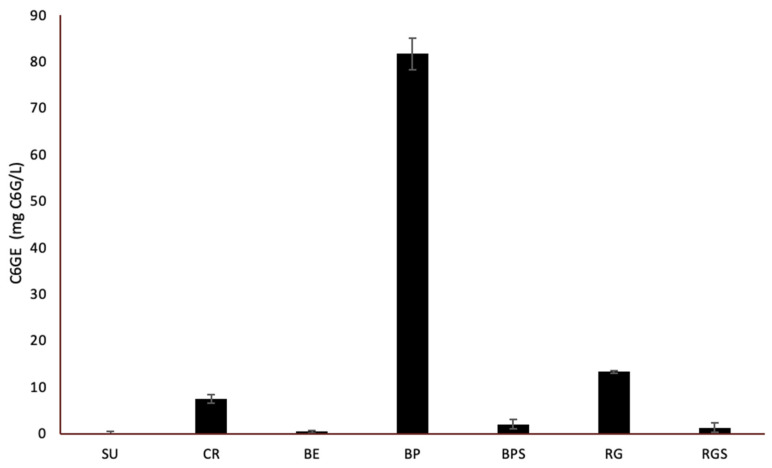
Quantification of Anthocyanins content in VVM pulp and seed extracts. The Graph shows the total anthocyanin content in the 7 extracts where it is expressed as cyanidin-6-O-glucoside equivalent (mg C6GE/L). The experiment was carried out in triplicates and repeated three times. Data are represented as mean values ± SEM.

**Figure 3 ijms-24-13440-f003:**
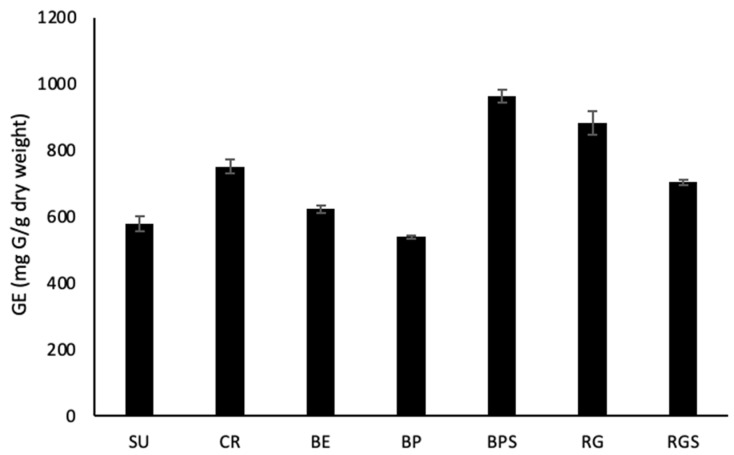
RSC (Reducing Sugar Content) in VVM pulp and seed extracts. A graph showing the total reducing sugar content in the different VVM extracts expressed as glucose equivalence per gram dry weight of extract (mg G/g dry weight. The experiment was carried out in triplicates and repeated three times. Data are represented as mean values ± SEM.

**Figure 4 ijms-24-13440-f004:**
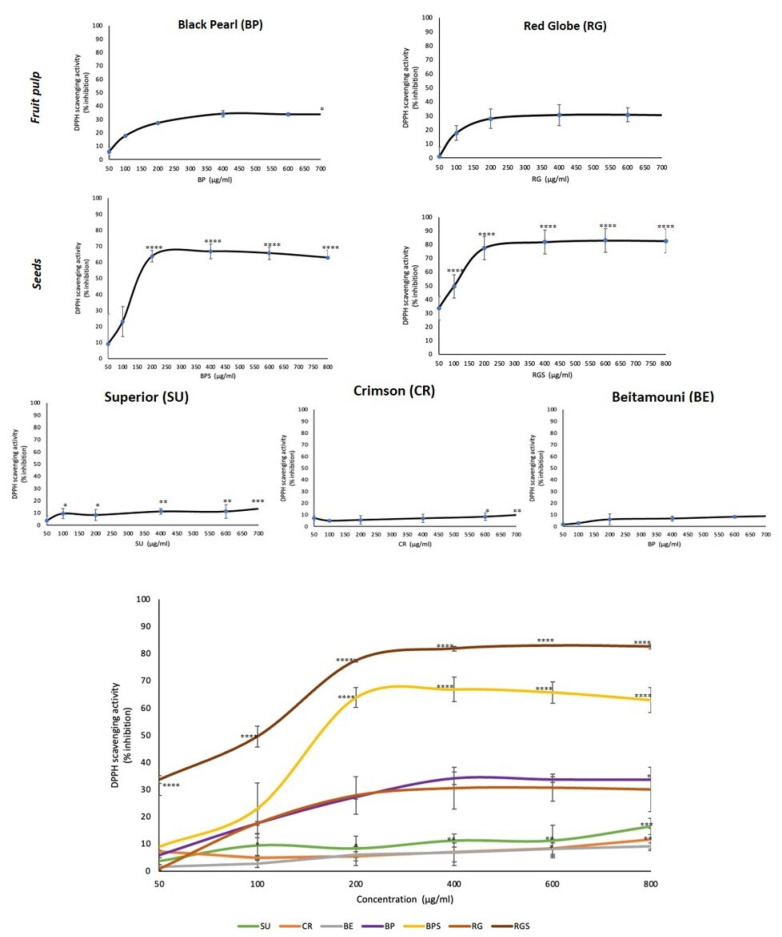
VVM extracts possess variable radical scavenging activities. The 7 graphs at the top show the DPPH radical scavenging activity in % as a function of the increasing concentration of VVM pulp and seed extracts in μg/mL whereas the last graph shows the DPPH radical scavenging activity as a function of the increasing concentration of all the 7 extracts in μg/mL. Values represent the means ± SEM of three independent experiments performed in triplicates and expressed as percentage of vehicle-treated control cells (* denotes *p* < 0.05, ** denotes a *p* < 0.005, *** denotes *p* < 0.001 and **** denotes *p* < 0.0001).

**Figure 5 ijms-24-13440-f005:**
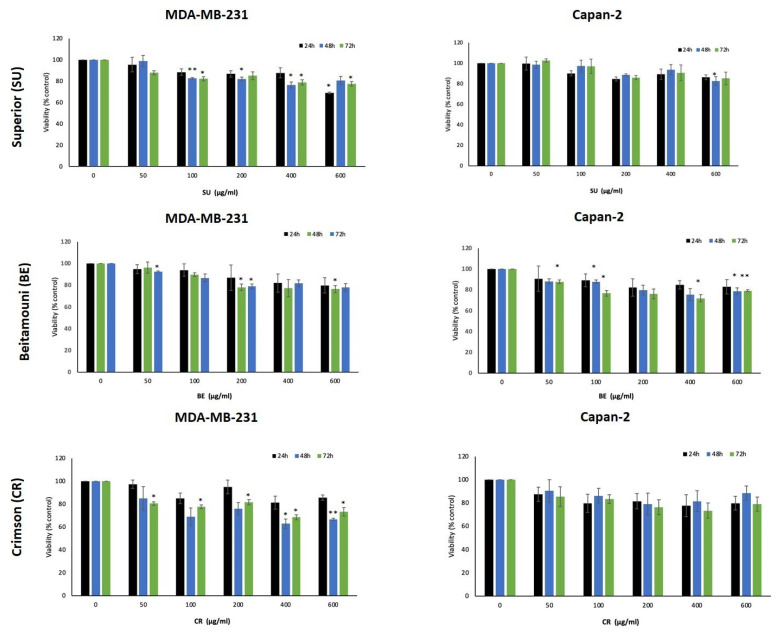
VVM extracts of the seedless varieties decrease the viability of Capan-2 pancreatic cancer and MDA-MB231 triple negative breast cancer. Capan-2 and MDA-MB231 cancer cells were treated with and without the indicated concentrations of VVM extracts for 24, 48, and 72 h. Cell viability was evaluated using the metabolic-dye-based MTT assay. Data represent the mean of three independent experiments performed in triplicate. Data represent the mean ± SEM of three independent experiments (*n* = 3). * denotes *p* < 0.05 and ** denotes a *p* < 0.005).

**Figure 6 ijms-24-13440-f006:**
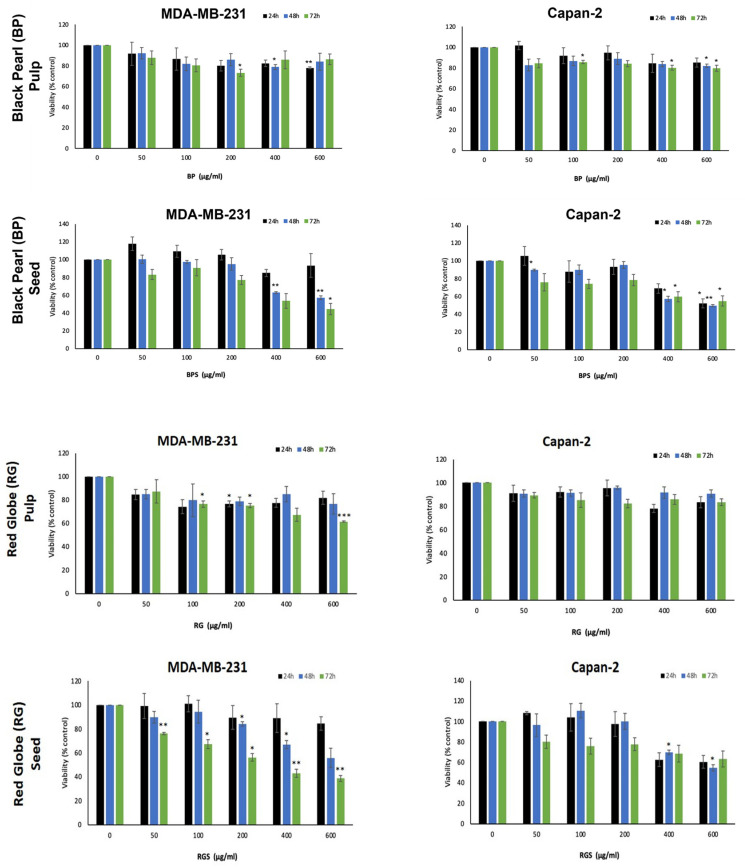
Grape pulp and seed extracts of the seeded varieties inhibit the proliferation of MDA-MB-231 breast cancer cells and Capan-2 pancreatic cancer cells. Capan-2 and MDA-MB231 cancer cells were treated with and without the indicated concentrations of VVM extracts for 24, 48 and 72 h. Cell viability was evaluated using the metabolic-dye-based MTT assay, as described in Materials and Methods. Data represent the mean of three independent experiments performed in triplicate. Data represent the mean ± SEM of three independent experiments (*n* = 3). * denotes *p* < 0.05, ** denotes a *p* < 0.005 and *** denotes *p* < 0.001.

**Table 1 ijms-24-13440-t001:** Phytochemical Analysis of VVM Pulp and Seed Extracts.

	SU	CR	BE	BP	BPS	RG	RGS
Tannins	-	-	-	-	+	-	+
Resins	-	-	-	-	-	-	+
Saponins	-	-	-	-	+	-	+
Phenols	+	+	+	+	+	+	+
Quinones	+	+	+	+	+	+	+
Steroids	-	-	-	-	-	-	-
Cardiac Glycosides	-	+	-	+	+	-	-
Terpenoids	+	+	+	+	+	+	+
Anthraquinones	-	-	-	-	-	-	-
Anthocyanins	+	+	+	+	+	+	+
Reducing sugar	+	+	+	+	+	+	+

**Table 2 ijms-24-13440-t002:** Phytochemical screening of Vitis vinifera pulp and seed extracts.

Phytoconstituents	Added Reagent	Expected Result
Reducing sugar	Drops of Fehling (A + B)	Brick-red precipitate
Anthraquinones	HCl (10%)	Precipitate
Tannins	Ferric chloride FeCl_3_ (10%)	Blue color
Resins	Acetone + water	Turbidity
Terpenoids	Chloroform + concentrated sulfuric acid	Reddish brown color on the surface
Quinones	HCl concentrated	Precipitate or yellow color
Sterols & steroids	Chloroform + concentrated sulfuric acid	Red color of the upper layer+ greenish yellow fluorescence in the acid layer
Anthocyanins	NaOH (10%)	Blue color
Cardiac glycosides	Acetic acid glacial + ferric chloride FeCl_3_ (5%) + concentrated sulfuric acid	Purple ring + Brown ring + Green ring
Saponins	Vigorous shaking	Layer of foam
Phenols	FeCl_3_ (1%) + K_3_(Fe(CN)_6_) (1%)	Greenish blue color

## Data Availability

All data included in the main text.

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
