# Peer review of "In Vitro Evaluation of Biological and Anticancer Activities Exhibited by Five Varieties of Vitis vinifera L."

_ijms, 2023, doi:10.3390/ijms241713440_

Round 1
Reviewer 1 Report
There are 9 missed citation to complete
Author Response
All citations are now completed.
Reviewer 2 Report
The article by Darwish et al. is a good intention to study the potential of the different varieties of V. vinifera on Capan-2 and MDA-MB231 cell lines.
Unfortunately, the article presents several writing flaws in addition to the lack of contribution of relevant or novel results. Some HPLC or mass analysis is recommended to enhance the importance of their different varieties on biological activities. Even for antioxidant activity, comparing them with at least two other colorimetric methods such as ABTS and FRAP, in addition to IC50 calculations is suggested. I started to review the entire document but it contains several errors in writing, grammar, punctuation, uniformity, and structure of ideas (it is suggested that you strictly adhere to the scientific publication format). Unfortunately, I could not continue with the observations due to the large number of errors on the part of the authors.
In general:
Remove intermediate lines in tables
Improve the resolution of the figures
Throughout the text authors use units such as ml or mL, please unify the style.
Throughout the text, the authors were not careful to write all scientific names in italics.
Line 2. Title: Write the scientific name in italics.
Line 87. The scientific name is not in italics but the word “extracts” is.
Line 88. Italics in the scientific name.
Line 88. The letters are in a different size. In addition, giving more conditions or characteristics of the collection place is suggested.
Line 96. Why did the authors use DMSO if it is supposed to be toxic? While for the phytochemical activity, I also saw that storage with this solvent is irrelevant when the extracts could be dissolved in solvents such as methanol or ethanol in the case of total phenolic content.
Line 97. Remove the colon from the title.
Line 102. Please check this error “Error! Reference source not found..”
Line 103. Is the qualitative identification of terpenoids and sterols the same reagent? Also for tannins, because no reagent other than ferric chloride was used if it is known to react with phenols...
Line 108. Remove the double parentheses.
Line 110. Why did the authors express the equivalents of gallic acid per mL if the extract was lyophilized and should be expressed per g of sample on a dry basis?
Line 110. “the Gallic acid standard curve 110 as Gallic acid equivalents (GAE) per mL of sample”. Irrelevant and sounds repetitive.
Line 131. In all the experimental sections, it is irrelevant to mention the following sentence “The experiment was carried out in triplicates and repeated three times. Data are represented as mean values ± SEM”. They can mention it in the statistical analysis section.
Line 133. Remove the colon at the end of the title. Also, didn't the authors have some reference standard like Trolox to compare the antioxidant activity?
Line 143. In the case of formulas, it is suggested that you use the equation editor.
Line 169. The authors say they use a significance level of 0.05 but in the graphs, they denote the significant differences by asterisks (*,**,***, etc.); they are asked to review the style.
Line 180. For tables 1 and 2, how do I differentiate the results for pulp and seed?
Line 182. Asserting “chemical composition” as a title seems delicate to me when the authors did not make or identify compounds by chromatographic techniques.
Line 194. Again this error appears “Error! Reference source not found..”
Line 198. It is confusing to choose units to express anthocyanins by L when it is expressed by milliliters for other activities.
Line 201. The graph does not show the significant differences between the treatments.
Line 207. “was quantified by measuring glucose equivalence per ml of extract (mg G/g dry weight” ??????
Line 228. The graph is messed up, they only mention the 7 graphs above, but what about the one below?
Line 239. It goes again “Error! Reference source not 239 found..” and then appears in the following sections!!!
Quite a few mistakes.
Line 274. But the tables are not differentiated to assert these results.
Line 354. Values for the correlation? You mean, r?
The language was difficult to read due to several grammatical mistakes.
Author Response
Remove intermediate lines in tables
The intermediate lines were removed
Improve the resolution of the figures
The resolution of all figures is now improved
Throughout the text authors use units such as ml or mL, please unify the style.
The unit ml is now used throughout the entire manuscript
Throughout the text, the authors were not careful to write all scientific names in italics.
All scientific names are now corrected
Line 2. Title: Write the scientific name in italics.
All scientific names have put in italics
Line 87. The scientific name is not in italics but the word “extracts” is.
The word extract is fixed
Line 88. Italics in the scientific name.
Applied as mentioned above
Line 88. The letters are in a different size. In addition, giving more conditions or characteristics of the collection place is suggested.
The letters' font has been adjusted for greater consistency, and a supplementary table has been included to provide further details regarding the vineyard location.
Line 96. Why did the authors use DMSO if it is supposed to be toxic? While for the phytochemical activity, I also saw that storage with this solvent is irrelevant when the extracts could be dissolved in solvents such as methanol or ethanol in the case of total phenolic content.
Following numerous trials, it was discovered that BPS and RGS are insoluble in methanol or ethanol. Therefore, DMSO was utilized instead. Additionally, the concentration of DMSO (volume/volume medium) did not exceed 0.1% in the highest tested concentration of VVM extract, ensuring its safety for cancer cells.
Line 97. Remove the colon from the title.
done
Line 102. Please check this error “Error! Reference source not found..”
This error is now fixed
Line 103. Is the qualitative identification of terpenoids and sterols the same reagent? Also for tannins, because no reagent other than ferric chloride was used if it is known to react with phenols...
The Salkowski test involves treating sterols or steroids with chloroform and concentrated sulfuric acid. When this reaction takes place, bluish red, cherry red, or purple colors appear in the chloroform layer. The test is performed by dissolving the extract samples in chloroform and mixing them with an equal volume of concentrated sulfuric acid. On the other hand, the presence of terpenoids is indicated by the formation of a reddish-brown color at the interface.
To detect tannins, Braymer's test is employed. In this test, ethanolic extracts of the plant parts are treated with 10% alcoholic FeCl3. The presence of tannins is confirmed when a blue-black color develops as a result.
For phenolic compounds, 0.5g of the plant extract is mixed with 5ml of ethanol and ultrasonicated for 15 minutes at 30 degrees Celsius. The mixture is then filtered, and 2 ml of distilled water is added to the filtrate. Subsequently, 10% FeCl3 is added, and the appearance of a dark green color signifies the presence of phenolic compounds.
(Source: https://www.scielo.br/j/bjb/a/X94rL4g8SPjCGTf89wKVBQG/?lang=en)
Line 108. Remove the double parentheses.
Removed
Line 110. Why did the authors express the equivalents of gallic acid per mL if the extract was lyophilized and should be expressed per g of sample on a dry basis?
The comparison involved assessing the radical scavenging activity of the extracts in relation to the control, thereby determining the percentage inhibition.
Line 110. “the Gallic acid standard curve 110 as Gallic acid equivalents (GAE) per mL of sample”. Irrelevant and sounds repetitive.
The unit is now adjusted
Line 131. In all the experimental sections, it is irrelevant to mention the following sentence “The experiment was carried out in triplicates and repeated three times. Data are represented as mean values ± SEM”. They can mention it in the statistical analysis section.
The paragraph is rephrased to meet the reviewer’s suggestion
Line 133. Remove the colon at the end of the title. Also, didn't the authors have some reference standard like Trolox to compare the antioxidant activity?
The title was revised by removing the colon at the end. The comparison focused on evaluating the radical scavenging activity of the extracts in comparison with the control, thereby effectively controlling the percentage inhibition.
Line 143. In the case of formulas, it is suggested that you use the equation editor.
The formulas are now added using the equation editor
Line 169. The authors say they use a significance level of 0.05 but in the graphs, they denote the significant differences by asterisks (*,**,***, etc.); they are asked to review the style.
All asterisks are explained in all figures’ captions and the statistical analysis part
Line 180. For tables 1 and 2, how do I differentiate the results for pulp and seed?
Table 1 presents the reagents used and the anticipated positive outcomes for each test. On the other hand, Table 2 displays the results of the tests conducted on seven prepared extracts: the pulp of red globe indicated as RG, the seeds of red globe denoted as RGS, the pulp of black pearl denoted as BP, and the seeds of black pearl denoted as BPS. Additionally, for the seedless varieties, SU stands for superior, CR stands for crimson, and BP stands for Beitamouni.
Line 182. Asserting “chemical composition” as a title seems delicate to me when the authors did not make or identify compounds by chromatographic techniques.
Chemical composition is now replaced by “ Quantitative Phytochemical Analysis”.
Line 194. Again this error appears “Error! Reference source not found..”
This error is now fixed
Line 198. It is confusing to choose units to express anthocyanins by L when it is expressed by milliliters for other activities.
Based on the manufacturer’s recommendation of the kit, this unit was chosen.
Line 201. The graph does not show the significant differences between the treatments
Statistical analysis is not possible due to the absence of anthocyanin, the violet pigment responsible for grape skin color, in white grape varieties..
Line 207. “was quantified by measuring glucose equivalence per ml of extract (mg G/g dry weight” ??????
This is rephrased as “measuring glucose equivalence per gram dry weight of extract (mg G/g dry weight”.
Line 228. The graph is messed up, they only mention the 7 graphs above, but what about the one below?
The one below is now described as follow:
whereas the last graph shows the DPPH radical scavenging activity as function of the increasing concentration of all the 7 extracts in μg/ml.
Line 239. It goes again “Error! Reference source not 239 found..” and then appears in the following sections!!!
This error is now fixed
Quite a few mistakes.
Line 274. But the tables are not differentiated to assert these results.
This point is a bit not clear, table 2 shows the different between the analyzed phytochemicals in all extracts.
Line 354. Values for the correlation? You mean, r?
This paragraph is rephrased to mention the correlation based on the principal component analysis (PCA).
Comments on the Quality of English Language
The language was difficult to read due to several grammatical mistakes.
The entire manuscript is edited and proofread.
Reviewer 3 Report
General comment
The manuscript ''In Vitro Evaluation of Biological and Anticancer Activities Exhibited by Five Varieties of Vitis Vinifera L.'' described the content of secondary metabolites (polyphenols, anthocyanins, reducing sugars) in fruits of five Vitis vinifera cultivars as well as their biological activity. Antioxidant activity was measured by DPPH and it was found that red cultivars possessed a higher antioxidant activity compared with the white ones. Results also shown that red verities have a potential anticancer activity against Capan-2 pancreatic cancer and MDA-MB231(TNBC) breast cancer cell lines. The activity was higher when the pulp was combined with the seeds.
In general, article should be better edit. At first, there are too many language mistakes and manuscript should be checked by professional linguistic.
Minor comments
Title
Line 2: change ''Vitis Vinifera'' into ''Vitis vinifera''
Abstract
Line 13: the names of the varieties should be writing on the botanical correct way through all manuscript parts. So, change ''Black Pearl (BP), Red Glob (RG), Crimson (CR), Beitamouni (BE) and Superior'' into ''V. vinifera 'Black Pearl' (BP), V. vinifera 'Red Glob' (RG), V. vinifera 'Crimson' (CR), V. vinifera 'Beitamouni' (BE) and V. vinifera 'Superior'
Line 15: omit abbreviation ''(BPS)'' because it was not mentioned again in the Abstract
Line 16: omit ''(RES)'' from the same reason
1. Introduction
Line 26: Line 2: change ''Vitis'' into ''Vitis vinifera L.''
Line 33: add space after ''worldwide''
Line 69: change ''Vinifera'' into ''vinifera''
Lines 82–84: write the names of cultivars on the botanical correct way
2. Materials and Methods
Line 87: write title of subsection ''Preparation of Vitis vinefera extracts'' according to journal style
Line 87: change ''vinefera'' into ''vinifera''
Line 88: change ''Vitis vinefera'' into ''Vitis vinifera''
Line 89: write the names of cultivars on the botanical correct way
Lines 91, 96: put a minus sign instead of a dash
Line 102: omit ''Error! Reference source not found.. ''
Line 103: put ''Vitis vinifera'' into Italic
Line 110: change ''the'' into ''The''
Line 115: put ''Vitis vinifera'' into Italic
Line 125: add space before ''mL''
Line 128: add space before ''nm''
Line 128: add space before ''0.5''
Line 153: put ''C'' into Normal
Line 156: change letter ''x'' with ''symbol ''×'''
3. Results
Line 170: omit colon
Line 175: shown in??
Thera are two Table 2, So, omit first Table 2 after line 175
Line 180: explain abbreviation VVM
Line 186: change letter ''x'' with ''symbol ''×'' after ''8.4'' and ''11''
Line 194: omit ''Error! Reference source not found.. ''
Figure 1: please, follow journal style when describe figures
Lines 198-199: omit ''Error! Reference source not found.. ''
Line 210: omit ''Error! Reference source not found.. ''
Line 212: explain abbreviations ''RSC''
Line 213: change ''ml'' with ''mL''
Lines 226, 239, 242, 244: omit ''Error! Reference source not found.. ''
Figure 4: please, follow journal style when describe figures. Graphs marked with SU, CR and BE show activity of pulp? Please, marked that. What is the difference among the last graph and others?
Line 253: omit ''as described in Materials and Methods''
4. Discussion:
Line 266: omit colon
Line 274: change ''Saponins'' into ''saponins'' (and through all text)
Line 274: change ''Tannins'' into ''Tannins''
Line 279: add space after ''cells''
Lines 280-281: change ''Phenols, Quinones, Terpenoids, Anthocyanins and Reducing'' into ''phenols, quinones, terpenoids, anthocyanins and reducing'' (and through all text)
Line 285: change ''we detected'' into ''it was detected'' (and through all text)
Line 288: change ''we found'' into ''it was found'' (and through all text)
Line 313: add space after ''stress''
Line 325: add space before ''(Lin et''
References
In general, authors should check the journal style for references and make some corrections. Some of problematic expressions are given below.
Article title should be writing on the same way. For example, compare reference no. 1 and no. 2, no. 14 and no. 15, and no. 22
Line 374: put ''Vitis vinifera'' into Italic
Line 382: omit comma after ''J.''? Check journal style.
Line 386: put ''Artemisia annua'' into Italic
Line 392, 401, 407, 430: put ''Vitis vinifera'' into Italic
Line 401: put ''Vitis vinifera'' into Italic
There are too many language mistakes and manuscript should be checked by professional linguistic
Author Response
Title
Line 2: change ''Vitis Vinifera'' into ''Vitis vinifera''
Done
Abstract
Line 13: the names of the varieties should be writing on the botanical correct way through all manuscript parts. So, change ''Black Pearl (BP), Red Glob (RG), Crimson (CR), Beitamouni (BE) and Superior'' into ''V. vinifera 'Black Pearl' (BP), V. vinifera 'Red Glob' (RG), V. vinifera 'Crimson' (CR), V. vinifera 'Beitamouni' (BE) and V. vinifera 'Superior'
Done
Line 15: omit abbreviation ''(BPS)'' because it was not mentioned again in the Abstract
Done
Line 16: omit ''(RES)'' from the same reason
Done
1. Introduction
Line 26: Line 2: change ''Vitis'' into ''Vitis vinifera L.''
Done
Line 33: add space after ''worldwide''
Done
Line 69: change ''Vinifera'' into ''vinifera''
Done
Lines 82–84: write the names of cultivars on the botanical correct way
Done
2. Materials and Methods
Line 87: write title of subsection ''Preparation of Vitis vinefera extracts'' according to journal style
Done
Line 87: change ''vinefera'' into ''vinifera''
Done
Line 88: change ''Vitis vinefera'' into ''Vitis vinifera''
Done
Line 89: write the names of cultivars on the botanical correct way
Done
Lines 91, 96: put a minus sign instead of a dash
Done
Line 102: omit ''Error! Reference source not found.. ''
This error is now fixed
Line 103: put ''Vitis vinifera'' into Italic
Done
Line 110: change ''the'' into ''The''
Done
Line 115: put ''Vitis vinifera'' into Italic
Done
Line 125: add space before ''mL''
Done
Line 128: add space before ''nm''
Done
Line 128: add space before ''0.5''
Done
Line 153: put ''C'' into Normal
Done
Line 156: change letter ''x'' with ''symbol ''×'''
Done
3. Results
Line 170: omit colon
Done
Line 175: shown in??
Thera are two Table 2, So, omit first Table 2 after line 175
Done
Line 180: explain abbreviation VVM
Done and explained as requested
Line 186: change letter ''x'' with ''symbol ''×'' after ''8.4'' and ''11''
Done
Line 194: omit ''Error! Reference source not found.. ''
This error is now fixed
Figure 1: please, follow journal style when describe figures
The description is fixed according to the journal style
Lines 198-199: omit ''Error! Reference source not found.. ''
This error is now fixed
Line 210: omit ''Error! Reference source not found.. ''
This error is now fixed
Line 212: explain abbreviations ''RSC''
Abbreviation explained as requested
Line 213: change ''ml'' with ''mL''
The unit is now unified as ml throughout the manuscript
Lines 226, 239, 242, 244: omit ''Error! Reference source not found.. ''
This error is now fixed
Figure 4: please, follow journal style when describe figures. Graphs marked with SU, CR and BE show activity of pulp? Please, marked that. What is the difference among the last graph and others?
The description is now corrected to follow the journal style. The part of the fruit used is described in the title of the figure where it’s noted that we are using the pulp of the seedless varieties only. The difference among the last graph and others is clarified in the title following your request
Line 253: omit ''as described in Materials and Methods''
Done
4. Discussion:
Line 266: omit colon
Done
Line 274: change ''Saponins'' into ''saponins'' (and through all text)
Done
Line 274: change ''Tannins'' into ''Tannins''
Done
Line 279: add space after ''cells''
Done
Lines 280-281: change ''Phenols, Quinones, Terpenoids, Anthocyanins and Reducing'' into ''phenols, quinones, terpenoids, anthocyanins and reducing'' (and through all text)
Done
Line 285: change ''we detected'' into ''it was detected'' (and through all text)
Done
Line 288: change ''we found'' into ''it was found'' (and through all text)
Done
Line 313: add space after ''stress''
Done
Line 325: add space before ''(Lin et''
The style of the references is changed
References
In general, authors should check the journal style for references and make some corrections. Some of problematic expressions are given below.
Article title should be writing on the same way. For example, compare reference no. 1 and no. 2, no. 14 and no. 15, and no. 22
The style of the references is changed based on the journal’s style and recommendation
Line 374: put ''Vitis vinifera'' into Italic
Done
Line 382: omit comma after ''J.''? Check journal style.
Done
Line 386: put ''Artemisia annua'' into Italic
Done
Line 392, 401, 407, 430: put ''Vitis vinifera'' into Italic
Done
Line 401: put ''Vitis vinifera'' into Italic
Done

Round 2
Reviewer 2 Report
Authors answered all questions mentioned in previous review. With this, the research that they conducted has better understanding. Only a few things should be corrected and they’re listed below. Thank you for considering the suggestions made.
Line 87: Vitis vinífera is not in italic.
Line 98: V. vinifera is not in italic.
Line 108: V. vinifera “Red Globe” is not in italic.
Line 308: It seems that sentence is not complete. I suppose that it should say “(VVM) extracts as shown in tables 1 and 2”. Also, name of table is missing.
There’s a lot of space between lines 313 and 319.
Author Response
Line 87: Vitis vinifera is not in italic.
Done
Line 98: V. vinifera is not in italic.
Done
Line 108: V. vinifera “Red Globe” is not in italic.
Done
Line 308: It seems that sentence is not complete. I suppose that it should say “(VVM) extracts as shown in tables 1 and 2”. Also, name of table is missing.
This sentence has been revised for clarity and conciseness
There’s a lot of space between lines 313 and 319.
The space is removed.